# A systematic review of barriers and enablers to South Asian women's attendance for asymptomatic screening of breast and cervical cancers in emigrant countries

Rachel Mary Anderson de Cuevas,[1] Pooja Saini,[2,3] Deborah Roberts,[4] Kinta Beaver,[5] Mysore Chandrashekar,[4] Anil Jain,[6] Eleanor Kotas,[1] Naheed Tahir,[1] Saiqa Ahmed,[1] Stephen L Brown[1]

RMAC and PS contributed equally.

For numbered affiliations see end of article.

**Correspondence to**
Dr Pooja Saini;
P.Saini@ljmu.ac.uk

## ABSTRACT

**Objectives** The aim of this review was to identify the cultural, social, structural and behavioural factors that influence asymptomatic breast and cervical cancer screening attendance in South Asian populations, in order to improve uptake and propose priorities for further research.

**Design** A systematic review of the literature for inductive, comparative, prospective and intervention studies. We searched the following databases: MEDLINE/In-Process, Web of Science, EMBASE, SCOPUS, CENTRAL, CDSR, CINAHL, PsycINFO and PsycARTICLES from database inception to 23 January 2018. The review included studies on the cultural, social, structural and behavioural factors that influence asymptomatic breast and cervical cancer screening attendance and cervical smear testing (Papanicolaou test) in South Asian populations and those published in the English language. The framework analysis method was used and themes were drawn out following the thematic analysis method.

**Settings** Asymptomatic breast or cervical screening.

**Participants** South Asian women, including Bangladeshi, Indian, Pakistani, Sri Lankan, Bhutanese, Maldivian and Nepali populations.

**Results** 51 included studies were published between 1991 and 2018. Sample sizes ranged from 25 to 38 733 and participants had a mean age of 18 to 83 years. Our review showed that South Asian women generally had lower screening rates than host country women. South Asian women had poorer knowledge of cancer and cancer prevention and experienced more barriers to screening. Cultural practices and assumptions influenced understandings of cancer and prevention, emphasising the importance of host country cultures and healthcare systems.

**Conclusions** High-quality research on screening attendance is required using prospective designs, where objectively validated attendance is predicted from cultural understandings, beliefs, norms and practices, thus informing policy on targeting relevant public health messages to the South Asian communities about screening for cancer.

### Strengths and limitations of this study

► Separate outcomes were compared of integrative reviews of inductive, predictive, comparative and intervention studies to assess consistencies between methods.
► Inductive studies provided nuanced and detailed insights into cultural, social, structural and behavioural factors influencing screening attendance.
► Deductive studies did not use insights gained from inductive research, were either atheoretical or used generic health psychology theories that were validated on Western samples and were generally poorly designed.
► Due to the small number of published studies, it is difficult to identify factors unique to groups of South Asian women based on nationality, geographical region or religion.
► We provide specific advice for high-quality deductive research on screening attendance that will allow estimation of the prevalence of factors that facilitate or inhibit screening attendance and the magnitude of their influence on attendance.

**PROSPERO registration number** CSD 42015025284.

## INTRODUCTION

Since 1945, many countries have benefited economically and socially from large-scale migration from the South Asian nations of India, Pakistan, Bangladesh, Sri Lanka, Nepal, Maldives and Bhutan.[1] Migration has largely favoured English-speaking countries, although large South Asian populations also exist in non-Anglophone European, African and neighbouring Asian countries.[2] In the UK, the South Asian population constitutes the largest ethnic minority category.[3] In all host countries, historic migration patterns

have led to the establishment of South Asian communities in cities and large towns where cultural norms and practices of the countries of origin are practised alongside those of the host country.[4]

In the UK, South Asian women have higher breast and cervical cancer mortality than the host population, worse cancer-related health outcomes, with the exception of some Indian groups, and are more likely to present with advanced disease.[3 5] While South Asian and host populations may differ over a range of factors that influence mortality, such as tumour subtype and human papillomavirus status (Gomez 2010), one potential cause of greater mortality is that South Asian women show a lower likelihood of attending routine mammographic and Papanicolaou (Pap) screening. Screening is widely available in most high-income countries.[6–8] Some research shows population mortality benefits of screening programmes,[9 10] although other studies find no effect.[11] Importantly, greater mortality benefits are found at the individual level, where studies confine analyses to women who accept screening rather than those who are merely invited (because some women decline screening).[12] Compared with the host population, South Asian women in England show lower uptake of breast screening services,[13–16] particularly those from lower socioeconomic groups[13 17 18] and a higher proportion have never received cervical screening.[19] This is also the case in the USA.[20]

Possible explanations for why screening rates are lower in South Asian populations have included poorer individual knowledge and awareness of breast and cervical cancer,[21–23] lower community awareness, poor communication between health professionals and patients, health professional background and less access to appropriate cancer health services.[24 25] Some South Asian women cannot speak or read in the host language.[26 27] Another body of research focuses on South Asian women's attitudes, beliefs and behaviours relating to cancer screening.[28 29] Crawford *et al*[29] and Sokal[26] recently compiled scoping and critical reviews of breast, cervical and colorectal screening in South Asian populations in Canada, the USA and the UK. The reviews demonstrate how individuals' beliefs, knowledge and perceptions of access barriers are shaped by the host environment, migration experience, cultural references and practices of the country of origin, and the cultural processes of adaptation to the host country.

Crawford *et al*'s and Sokal's reviews have limitations. Both, combined studies with differing methodological approaches to achieve integrated descriptions of findings. This approach provides a comprehensive overview but may lead to interpretation bias because it does not separate content from method.[26] This leads to two limitations. First, critical examination of study quality is more difficult when varying methodologies are used. Thus, the value of findings cannot be easily moderated or weighted by quality appraisal of the reviewed studies. Second, it is important that findings are replicated across methods. For example, inductive research permits detailed phenomenological understandings of factors that facilitate or inhibit screening, but not epidemiological estimates of the prevalence of these factors or the magnitude of their influence on screening. This requires well-designed quantitative studies.[30] Similarly, quantitative research alone is unlikely to be sensitive to local complexities unless complemented by inductive approaches. When these approaches are conflated, as with Crawford *et al*'s and Sokal's approaches, the reader cannot determine if insights are or are not replicated across different approaches.

Whittemore and Knafl[31] describe a method of integrative review that resolves these problems by separately integrating findings within different methodologies, then comparing the integrative findings across analyses to identify consistencies and limitations of findings within and between methods. Researchers into migrant health use four basic types of investigation. *Inductive* studies use qualitative analyses that allow participants to present their own experiences, thus providing novel insights that drive theory development. *Predictive, comparative* and *intervention* studies are deductive, using quantitative methods to test hypotheses. *Predictive* studies predict health behaviour from measures of individual and contextual attributes, allowing theory testing by quantifying associations between predictors and outcomes within migrant populations. *Comparative* studies compare target populations with host or other immigrant populations to identify whether the determinants of health behaviour in immigrant groups are unique to them or are shared with host or other immigrant groups. Shared factors include relative economic deprivation[26 32] or social and cultural adjustment challenges.[33] It is also important to review reports of *intervention* studies to examine how successful previous interventions (or their individual components) have been in improving screening rates in South Asian populations.

## Aims of the review

We examined cultural, social, structural and behavioural factors that influence asymptomatic breast and cervical cancer screening attendance in South Asian populations, to explain why attendance rates are lower than host country women. We performed separate integrative reviews of inductive, predictive, comparative and intervention studies, and compared outcomes of these reviews to assess consistencies between methods. Our aim was to identify the cultural, social, structural and behavioural factors that influence asymptomatic breast and cervical cancer screening rates in South Asian populations to improve screening rates and to propose priorities for further research. Our objectives were to:

► Critically review and integrate findings of inductive, predictive, comparative and intervention studies on asymptomatic screening.
► Document consistent and inconsistent findings across methods, and make theoretical and methodological recommendations for the conduct of future research.

## METHODOLOGY
### Search strategy
We conducted literature searches using multiple databases to overcome problems associated with inadequate indexing[31 34] and to ensure a more exhaustive scope.[31 35 36] We searched the following databases: MEDLINE/In-Process, Web of Science, EMBASE, SCOPUS, CENTRAL, CDSR, CINAHL, PsycINFO and PsycARTICLES for four key concepts: (1) South Asian population, (2) cancer, (3) asymptomatic breast or cervical screening and (4) knowledge, attitude, practice, behaviour or compliance. PubMed was searched for publications ahead of print and conference proceedings. Search terms were revised after initial searches revealed new terms. MeSH terms were run in combination with free-text searches of titles and abstracts. These are available as an online data supplement at https://www.crd.york.ac.uk/PROSPERO-FILES/25284_STRATEGY_20170702.pdf. Searches were conducted from database inception to 23 January 2018. The search was restricted to original research in English for all publication dates. Citations of selected studies were reviewed to identify any additional studies. We checked for grey literature via databases and repositories such as Open SIGL, Open Grey, PsycEXTRA, HMIC UK, The Grey Literature Report, ClinicalTrials.gov, National Technical Information Services (NTIS), National Cancer Intelligence Network (NCIN) and the WHO International Clinical Trials Registry Platform, and cancer and clinical networks including American Cancer Society, South Asian Health Foundation and MacMillan Cancer Support.

### Selection criteria
The review included studies on the cultural, social, structural and behavioural factors that influence asymptomatic breast and cervical cancer screening attendance and cervical smear testing (Pap test) in South Asian populations. It was confined to host countries where mass screening programmes are available to the general public, including South Asian Women. The populations of interest were Bangladeshi, Indian, Pakistani, Sri Lankan, Bhutanese, Maldivian and Nepali populations (or ethnic subgroups thereof). To ensure that content was not confounded by inclusion of other groups, studies needed to report on samples or subsamples identifiable as wholly South Asian, meaning that we accepted papers that examined South Asian and other samples provided that authors explicitly specified where South Asian content differed from other samples (inductive studies) or where South Asian samples were analysed separately or were specifically identified in moderation analyses (predictive, comparative or intervention studies).

To ensure that the studies pertained to screening attendance, we excluded those that did not specifically refer to screening. Thus, studies solely covering general attitudes to breast or cervical cancer were excluded. The review did not include breast self-examination, diagnostic screening or visual or tactile examinations by healthcare professionals. We excluded studies of women in known high-risk groups who were engaged in monitoring programmes for genetic risk factors, hereditary breast and ovarian cancer syndrome, premenopausal or familial breast cancer. We excluded homogeneous samples restricted to particular demographic groups because these are not population representative (eg, a study of dental students).

### Screening
Team members screened titles and abstracts to identify potentially eligible studies and two reviewers independently considered the eligibility of each of the titles and abstracts. Outputs were compared with detect discrepancies and the agreement rate was 90%. Disagreements over selection of abstracts were resolved by consensus between the team. Calibration of the selection criteria was performed after the first 50 and 100 papers and taking a small sample (15%) of reports from grey and unpublished literature. Two reviewers independently assessed the full text of relevant studies using a standardised, pilot-tested screening form agreed with the steering group. Disagreements were resolved by consensus or by referral to a third-party arbiter. EndNote (X5) reference manager was used to manage citations and view abstracts and full-text articles.

### Quality evaluation
Each study was evaluated for quality specific to the method used, with validated checklists developed from the Critical Appraisal Skills Programme.[37] Inductive studies were generally found to be good. Predictive, descriptive and intervention studies had theoretical, sampling, design and measurement limitations. We did not exclude studies that used poor methodologies, but extensively describe these problems and consequent interpretive limitations in the results.

### Data extraction, synthesis and analysis
All studies included in the review are included in summary tables (tables 1–4). Four reviewers completed data extraction for each study type and reviewed the variable headings on completion.[38] Subsequently, tables were adapted and the following variables were recorded for all studies: region, study design (sample size and sampling), demographic and clinical characteristics of women selected, setting, data collection instruments, analytical method, nature of asymptomatic screening (mammogram or Pap smear test), definition of timely screening attendance, theoretical focus, key findings, study limitations and quality rating. For predictive studies, we recorded outcome variables, rate of screening attendance and all predictors for and against screening. Intervention studies included a description of the intervention concerned.

Syntheses were made using thematic analysis within each methodology type.[31 39] Syntheses were initially structured from the summary tables, beginning with a period of data familiarisation, during which researchers listed ideas about emerging themes which formed the basis of a thematic framework. At this point, the analysis returned to the full papers, where the developing thematic

**Table 1** Inductive studies

| Region | Sample size | Sampling frame | Focus | Findings |
|---|---|---|---|---|
| **Breast cancer** | | | | |
| *Ahmad et al[47] (2012)* | | | | |
| Toronto Canada | 60 Indian and Pakistani immigrant women, 50+years; never screened or screened>3 years ago. | Concept mapping. Clustering of participant-generated statements. | Experiences and beliefs concerning barriers to mammography | Barriers to regular screening mammogram: lack of knowledge; fear of cancer and language and transportation. Barriers differed significantly according to years lived in Canada: dependence on family; ease of access to mammogram centre; language and transportation; fear of cancer and self-care. |
| *Bottorff et al[41] (1998)* | | | | |
| Large urban setting, Western Canada | 50 SA women, 30+years. FGDs with 30 mostly new informants. | IDIs and FGDs with healthy immigrant SA women via SA investigators' networks. | Beliefs attitudes and values related to breast health practices and screening | Beliefs centred on four domains: (1) A woman's calling—keeping the family honour, modesty and putting others first; (2) beliefs about cancer; (3) taking care of your breasts; (4) accessing services. |
| *Meana et al[42] (2001)* | | | | |
| Toronto, Canada | 30 recently immigrated Tamil women from Sri Lanka≥50 years. | Members of a SA Women's Centre. Three FGDs. | Attitudes/ beliefs regarding BC and BC screening | Common barriers to BC screening: (1) lack of understanding of the role of early detection in medical care; (2) religious beliefs; (3) fear of social stigmatisation. Other barriers: embarrassment about mammography procedures. No reported opposition from husbands. |
| *Pons-Vigués et al[43] (2012)* | | | | |
| Barcelona, Spain | 68 healthy women (6 Pakistani, Indian women), 40–69 years. | Key informants, cultural mediators and associations. | Concept of health prevention and knowledge, perceived benefits/barriers | Health prevention concept lay across three axes: (1) *understanding of prevention*; (2) *proactive or deterministic conception of health disease*; (3) women cared little for their own health but obliged to others. |
| **Cervical cancer** | | | | |
| *Bottorff et al[44] (2001)* | | | | |
| Western Canada | 20 SA (Sikh, Hindu, Muslim) women; 20+years, had Pap test. | IDIs with SA women attending for Pap testing organised by ethnic group. | Experiences and views concerning testing, their expectations and preferences | Perceptions of Pap testing: uncertainty about benefits of early detection in the absence of symptoms; reservations about screening unmarried young women due to preserving virginity; seen as beneficial to keep healthy and protect families from disease. Interplay between cultural values and healthcare system structures: shyness and discomfort discussing Pap test with physician. |
| *Haworth et al[45] (2014)* | | | | |
| Nebraska mid-Western USA | 27 healthy Bhutanese refugee women; 19–60 years. | Snowball sample community venues and residences (two FGDs). | CC and screening knowledge; susceptibility severity of CC; benefits/ barriers to screening | Most women had never heard of CC (or HPV) and felt it did not occur in their community. Women not familiar with concept of health prevention. Barriers: shyness; feelings of exposure and potential stigma; historical abuse, sexual assault and inappropriate behaviour by male HCPs in refugee camps; language; navigating a complex health system; limited insurance coverage; transportation; male translators. |
| *Oelke and Vollman[46] (2007)* | | | | |
| Urban Canada | 53 immigrant Sikh women, 21 to 65+years. Residency in Canada 6 months—32 years | Community locations, key contacts and Punjabi radio (13 IDIs). Community agency and English classes (3 FGDs). | Knowledge, understanding and perceptions of CC screening | *'Inside/outside'*: difficult to move ' outside' into Canadian society. *Individual*: unaware of importance of prevention; cervix as unknown body part; SRH not discussed. *Knowledge*: minimal knowledge of Pap test and no ready access to information. *Prevention*: not necessary in absence of symptoms. *Family*: cultural constraints; domination by males/elders; needing permission for medical appointments; a woman's sacrifice for the family. *Community*: preserving honour/status; shame surrounding inappropriate topic. *Healthcare system*: sex of physician; language barriers; trust; confidentiality and dearth of acceptable HCPs. |
| **Breast and cervical cancer** | | | | |
| *Hulme et al[49] (2016)* | | | | |
| Canada | 20 Bangladeshi women (12 individual interviews, 8 in focus groups), 30–65 years, residency in Canada; 7 women <5 years, 7 women ≥5 years, 6 women NA. | Selected from participants at a community-based education programme. | Knowledge, perceptions of barriers, role of family physicians and preferences for future access | Risk perception associated with personal experience, screening poorly understood in absence of symptoms; language barriers important; role of family physicians important, particularly females (who administer) cervical screening; fear of cancer inhibits screening; importance of self-efficacy, particularly in how self-efficacy is reflected in personal identity. |

BC, breast cancer; CC, cervical cancer; FGD, focus group discussions; HCP, healthcare provider or professional; HPV, human papillomavirus; IDI, in-depth interviews; Pap test, Papanicolaou test; SA, South Asian; SRH, sexual and reproductive health; NA, never attended.

framework was tested and refined against the initial data. Themes were developed, reviewed and refined by analysing the data synthesised within each code and testing for 'internal homogeneity' and 'external heterogeneity'.[40] The research group met continuously to check and discuss the meaning and interpretation of the data.

**Table 2**  Predictive studies

| Region | Sample size | Sampling frame | Focus | Outcome variable | Rate | Risk factors for not screening |
|---|---|---|---|---|---|---|
| **Breast Cancer Study** | | | | | | |
| Ahmed and Stewart[63] (2004) | | | | | | |
| Canada | Cross-section 54 SA women, aged 18+ years, Hindu or Urdu speakers | Attendees family medical clinic | HBM | Ever had CBE | 38.5% | Younger age, more barriers |
| Boxwala et al[83] (2010) | | | | | | |
| Detroit, USA | Cross-section 160 Indian women, 39+ years | Cultural or religious locations | HBM | Mammogram and CBE within 2 years | 63.8% | Not graduate education, disagree screening useful, mammogram less relatively important, not recommended by HCP |
| Chawla et al[57] (2015) | | | | | | |
| California, USA | Cross-section 186 SA women aged 50–74 years | Random digit telephone survey | None | Mammography test within 2 years | 79.5% | Not married, <25% of lifetime in the USA, no physician visits in last year |
| Hasnain et al[61] (2014) | | | | | | |
| Chicago, USA | Cross-section 105 SA first-generation Muslim women | Snowball sample | Anderson Behavioural Model for Health Service Use Anderson model), HBM, Transtheoretical model | Mammography test within 2 years (adherent), mammography test not within 2 years (overdue), never screened | 41% adherent, | Fewer years in the USA, lower mammogram importance, more barriers, lower intention |
| Islam et al[84] (2006) | | | | | | |
| New York, USA | Cross-section 43 women 18+ years | Attendees at cultural events | None | Mammogram test within 2 years | 55.8% | Uninsured, >10 years living in the USA |
| Kwok et al (2015) | | | | | | |
| Sydney, Australia | Cross-section 242 women 18+ years born in India or in Indian communities | Attendees at cultural events | Culture-specific factors | Have biannual mammography | 17.8% | Less time in Australia, divorced, separated, widowed |
| Marfani et al[62] (2013) | | | | | | |
| Baltimore, USA | Cross-section 418 Indian women | Attendees at cultural and religious events | Acculturation | Mammography or CBE within 1 year, Mammography or CBE within 2 years, mammography or CBE >2 years ago | Not provided | Low self and outcome efficacy for screening, greater barriers, lower acculturation, lower acculturation interacting with greater anxiety about BC |
| Meana et al[42] (2001) | | | | | | |
| Canada | Cross-section 122 Tamil women, 50+ years | Attendees at community and religious centres | HBM | Had mammograms | 57.4% | Fewer years in North America, more barriers |
| Menon et al[65] (2012) | | | | | | |
| Chicago, USA | Cross-section 330 SA women 40+ years | Community-based agencies | Precede-Proceed model | Ever had mammography | 65.5% | Less than 5 years in the USA, greater barriers, lower English language preference, never had cervical screening |
| Misra et al (2011) | | | | | | |
| USA cities | Cross-section 389 Indian women 40+ years | Random survey | None | Ever had mammography | 81.2% | Fewer years in the USA, No health insurance |
| Misra et al (2011) | | | | | | |
| USA cities | Cross-section 519 Indian women 18+ years | Random survey | None | Ever had Pap test | 74.2% | Fewer years in the USA, Lower education, no health insurance, no family cancer history |
| Pourat et al (2010) | | | | | | |
| California, USA | Cross-section 134 SA women 40+ years | Random survey | Acculturation | Mammogram within 2 years | 39% | None |
| Vahabi et al (2016) | | | | | | |

**Table 2** Continued

| Region | Sample size | Sampling frame | Focus | Outcome variable | Rate | Risk factors for not screening |
|---|---|---|---|---|---|---|
| Ontario, Canada | 18880 women aged 50–69 years | Government. Database linkage study | None | Verified mammography attendance within 2 years | 63.7% | Fewer years in Canada, no general GP assessment, GP trained overseas |
| Vahabi et al[53] (2017) | | | | | | |
| Ontario, Canada | 14352 women aged 50–74 years | Government. Database linkage study | Muslim majority country of origin | Verified mammography attendance within 2 years | 44.02% Muslim majority country, 45.41% non-Muslim majority | Muslim majority country of origin, male family doctor, family class immigrant, not speaking English and French, fee-for-service primary care or no primary care |
| **Cervical Screen Study** | | | | | | |
| Chaudhry et al (2003) | | | | | | |
| USA | Cross-section 225 SA women aged 15–83 years | SA family names | Anderson model | Pap test within 3 years | 73% | Unmarried, no bachelor degree, no usual source of medical care, <25% of lifetime in the USA |
| Chawla et al[57] (2015) | | | | | | |
| California, USA | Cross-section 711 SA women aged 21–74 years | Random digit telephone survey | None | Pap test within 3 years | 79.5 | Younger age, not married,<25% of lifetime in the USA |
| Gupta et al (2002) | | | | | | |
| Toronto, Canada | Cross-section 62 SA university students, 62 Tamil women aged 18–60 years | Common areas of university, Tamil community centres | Acculturation | Ever had Pap test | 25% | Lower education, education outside Canada, lower acculturation |
| Islam et al[84] (2006) | | | | | | |
| New York, USA | Cross-section 98 women 18+ years | Attendees at cultural events | None | Pap test within 3 years | 54.4% | Tested within 3 years: lower education, lower income, uninsured, <10 years living in the USA |
| Kue et al[66] (2017) | | | | | | |
| Columbus, Ohio, USA | Cross-section 97 Bhutanese-Nepali refugees 18+ years | Convenience sample at community locations | Beliefs, barriers and postmigration difficulties | Ever had Pap test | 44.3% | No positive perceptions of test, greater barriers, not recommended by HCP family or friends, fewer postmigration difficulties |
| Lin et al[58] (2009) | | | | | | |
| California, USA | Cross-section 338 SA women 18–65 years | Random telephone survey | None | Pap test in last 3 years | 73% | Not married, low income, no usual source of medical care |
| Lofters et al[54] (2017) | | | | | | |
| Canada | | Government. Database linkage study | Muslim majority country of origin | Verified Pap test in last 3 years | | Muslim majority country of origin, lowest income male family doctor. Family doctor not Canadian graduate, family class immigrant, not speaking French, fee-for-service primary care or no primary care |
| Marlow et al (2017) | | | | | | |
| UK | Cross-section of 230 SA women | Cluster randomised community survey of UK addresses | Precaution Adoption Process Model | Four group classification; unaware, unengaged, undecided, intention to be screened | 79% | |
| Menon et al[65] (2012) | | | | | | |
| Chicago, USA | Cross-section 330 SA women 40+ years | Community-based agencies | Precede-Proceed model | Ever had cervical screen | 32.8% | Lower education, greater barriers, lower English language preference, never had mammogram |

Continued

**Table 2** Continued

| Region | Sample size | Sampling frame | Focus | Outcome variable | Rate | Risk factors for not screening |
|---|---|---|---|---|---|---|
| Misra *et al* (2011) | | | | | | |
| USA cities | Cross-section 519 Indian women 18+ years | Random survey | None | Ever had Pap test | 74.2% | Fewer years in the USA, lower education, no health insurance, no family cancer history |
| Pourat *et al* (2010) | | | | | | |
| California, USA | Cross-section 195 SA women 40+ years | Random survey | Acculturation | Pap test within 3 years | 73% | Greater distance to Asian clinic, no health insurance, no private doctor, has previously delayed obtaining medical care, has had problem obtaining satisfactory doctor over past year |

BC, breast cancer; CC, cervical cancer; FGD, focus group discussions; GP, general practitioner; HBM, health belief model; HCP, healthcare provider or professional; IDI, in-depth interviews; Pap, Papanicolaou; SA, South Asian; SRH, sexual and reproductive health; CBE, clinical breast examination.

## Patient and public involvement

The research question was derived following the author (PS) attending community intervention sessions with South Asian women. There, the lack of knowledge of female cancers and the stigma associated with female cancer became apparent. Following some discussions, the group were asked about their own experiences and whether they would like to be part of future research to gain more understanding of the cultural, social, structural and behavioural factors that influence breast and cervical cancer screening attendance in South Asian populations. The group of attendees at the community sessions were invited to be involved in a funding application being submitted to the NIHR Collaboration for Leadership in Applied Health Research and Care North West Coast (CLAHRC NWC) and then the research group if the funding was awarded. Two women from the community (NT and SA) were interested then invited to join the research team made up of academics and clinicians. They were then signed up as NIHR CLAHRC NWC Public Advisors. All team members were involved in reviewing the submitted grant application and subsequently attended all steering group meetings where the search terms were finalised for the systematic review. The researchers, a seconded nurse from the local hospital and public advisors attended all training associated with conducting a systematic review, reviewed titles, abstracts and full papers for inclusion and exclusion and attended data analysis meetings. The public advisors and main researcher have disseminated the preliminary study findings at national and regional conferences, national meetings, community public engagement events and at the University of Liverpool. Both public advisors have become active members of the wider NIHR CLAHRC NWC structure since joining this review project and other women from the same community are now involved in other studies across the area. Their contribution has been invaluable.

## RESULTS

The combined search of electronic bibliographic databases yielded 10 969 citations (figure 1). After removing duplicates (n=3714), the remaining 7255 were screened on title and 1136 on abstract and 132 records were selected for full-text review. Subsequently, 81 were excluded on full text and 51 met the criteria for inclusion in the review. The 51 studies were published between 1991 and 2017 and were conducted in the USA (n=22), Canada (n=16), UK (n=5), Spain (n=2), Singapore (n=2), Malaysia (n=2), Hong Kong (n=1) and Australia (n=1). Sample sizes ranged from 25 to 38 733. Participants were recruited from community and healthcare settings and had a mean age of 18 to 83 years. Of 51 studies, 8 were inductive (see table 1), 25 predictive (containing analysis of predictors of and risk factors for attendance) (see table 2), 10 comparative (see table 3), and 8 intervention studies (see table 4). No further studies were found from the grey literature search.

### Overview

Inductive studies provided rich insights into cultural practices and assumptions, and the problems of adjusting to a new social and healthcare system that might inhibit screening in South Asian women. Largely, though, deductive studies failed to exploit these insights in hypothesis testing. Deductive studies were either atheoretical or used generic health psychology theories, such as the health belief model (HBM), that were validated on Western samples and not adapted for South Asian populations.

Nonetheless, common findings emerged across methodologies. The extent to which women understood the causes of cancer and the benefits of screening was important. Inductive studies revealed cultural constraints on understanding, while comparative studies showed South Asian women faring worse on measures of knowledge than host country women. Predictive studies showed that those with more complete understandings of cancer and screening were more likely to attend screening. Similarly, both inductive and deductive studies showed that

**Table 3** Comparative studies

| Region | Sample size | Sampling frame | Focus | Findings |
|---|---|---|---|---|
| **Breast cancer** | | | | |
| Abdul Hadi et al[70] (2010) | | | | |
| Penang State, Malaysia | 65 healthy Indian women aged>15 years, 177 Malay and 121 Chinese | Two shopping malls | Differences in knowledge/ perception of BC | Indians have less knowledge about risk factors, symptoms and screening options (subsidised mammography and CBE) compared with Malay and Chinese. Univariate analysis confounded by Indian population being least educated. |
| Pons-Vigués et al[43] (2012) | | | | |
| Barcelona city, Spain | 25 Pakistani–Indian women 45–69 years, 275 Spanish women 660 other immigrant groups | Sampled from Census respondents | Adapted HBM based on qualitative pilot study (Pons-Vigues et al[43] (2012)) | Indian–Pakistani women perceived more barriers to mammography screening than host country women, but fewer than other immigrant groups. |
| Sim et al (2009) | | | | |
| Singapore | 80 Indian women, 182 Malay, 700 Chinese, 38 other | Visitors to general hospital (not patients) | Knowledge and beliefs about BC and screening practices | No differences between Indian women and others in either knowledge or having ever attended a screening mammogram. |
| Teo et al[67] (2013) | | | | |
| Singapore | 52 locally raised Indian women, 104 Chinese, 52 Malay | Female patients and visitors to polyclinic, aged 40–70 years | No theoretical model | Indian women less likely to have ever had mammogram compared with majority Chinese, but more likely than Malays. Indian women *least* likely group to cite cost or potential pain as barriers to attending mammography. |
| Vahabi et al (2016) | | | | |
| Ontario, Canada | 18880 South Asian, 85872 other immigrant groups | Government database linkage study | No theoretical model | Lower mammography attendance in previous 2 years than other immigrant groups. |
| Wu et al[68] (2006) | | | | |
| Michigan, USA | 38 Indian women aged ≥40 years, X Chinese, X Filipino | Community or religious groups; ethnic student associations, community events | HBM | No difference in CBE and mammography take up between ethnic groups. Indian women had lower scores on perceived susceptibility and seriousness than Filipino and Chinese controlling income. Indian women more likely to say 'do not know where to find mammogram'. |
| Wu et al[69] (2008) | | | | |
| Michigan, USA | 109 Asian Indians aged≥40 years, literate | Community events, cultural centres, faith-based organisations, Asian health fairs | HBM | No group differences. |
| **Cervical cancer** | | | | |
| Dunn and Tan[55] (2010) | | | | |
| Malaysia | 96 married Indian women aged 25–65 years | Two-stage stratified-cluster random sampling | No theoretical model | Ever had Pap test: Indian population least likely to have ever had screening. Indian women who had ever received screening less likely to know its purpose than Malays. Indian women who had never had Pap test were 9% less likely to cite 'embarrassed' as reason for not undergoing testing. |
| Marlow et al[33] (2015) | | | | |
| England, UK | 120 Indian, 120 Pakistani, 120 Bangladeshi women, 120 white British, 120 Caribbean and 120 African | Quota sampling, random sampling within high ethnic concentration postcodes | No theoretical model | Indian, Pakistani and Bangladeshi women less likely to be screened over last 5 years than white British. Less knowledge than white British. |
| So et al[59] (2017) | | | | |
| Hong Kong | 161 Indian, Nepali and Pakistani women, 959 Chinese women, 50+ years | Community centres or associations, Chinese sample recruited using random digit dialling | No theoretical model | SA women less likely to have been screened, had fewer tests in previous 6 years, longer time since last test. |

BC, breast cancer; CC, cervical cancer; FGD, focus group discussions; HBM, health belief model; HCP, healthcare provider or professional; IDI, in-depth interviews; Pap, Papanicolaou; SA, South Asian; SRH, sexual and reproductive health.

perceived barriers inhibited screening and that South Asian women typically perceived more and different barriers to host country women. Inductive studies showed the cultural origins of barriers, describing how traditional beliefs about risk, illness, female roles and family structures mitigated screening interest and attendance. Predictive studies showed that the number of perceived barriers inhibited screening and that South Asian women

**Table 4** Intervention studies

| Region | Sample size and sampling frame | Intervention | Focus | Findings |
|---|---|---|---|---|
| **Breast cancer** | | | | |
| Ahmad et al[71] (2005) | | | | |
| Toronto, Canada | 127 SA immigrant women. Mean age 37 years (SD 9.7); lived 6 years in Canada (SD 6.6). n=82 preintervention; n=74 postintervention. | Pre (Prl)–Post (Pol) intervention comprising written socioculturally tailored language-specific health education materials. | Barriers to mammography screening HBM, Stages of Change model). | A significant increase in self-reporting 'ever had' routine physical check-up (46.4%–70.8%; p<0.01) and CBE (33.3%–59.7%; p<0.001). Decrease in: misperception of low susceptibility to women with BC (3.0–2.4; p<0.001); misperception of short survival after diagnosis (2.7–1.8; p<0.001); and perceived barriers to CBE (2.5–2.1; p<0.001). Self-efficacy to have CBE increased (3.1–3.6; p<0.001). |
| Hoare et al[75] (1994) | | | | |
| Oldham, UK | 5277 women with SA names from general practices with high number of SA patients. Pakistani/intervention n=145 (59%); Bangladeshi/controls n=87 (57%). | RCT: 527 women stratified into Pakistani (n=324) intervention and Bangladeshi (n=203) control groups. | Awareness of screening. | No difference in attendance was found between the intervention and control groups (49% and 47%). Attendance for screening was related to length of stay in the UK. |
| Sadler et al[73] (2003) | | | | |
| San Diego County, USA | Asian and Pacific Islander women from San Diego County. Indian n=125. Women aged>20 years (screening from 20 onwards). | Preintervention and postintervention. The Asian Grocery Store based cancer intervention programme, incorporating an educational programme into women's routine shopping activities. | Barriers to mammography screening. | Shift towards screening uptake for Chinese and Vietnamese American women who were non-adherent at baseline but no change for Asian Indian and Japanese American women at follow-up. |
| **Cervical cancer** | | | | |
| Grewal et al[76] (2004) | | | | |
| Vancouver, Canada | Specialised Pap test clinic for SA women. 1995–1998; 61–107 – 35 new visits in the intervention. Reasons for non-attendance n=74. | Time series of service use. Community initiative led by SA community health nurses in collaboration with influential women in the SA community, local physicians and health board authorities. Qualitative interviews with 20 women who attended the Pap test clinic. | Awareness of screening. | Attendance patterns were not maintained although women had positive experiences. Challenges for ongoing success: (1) maintaining the continued involvement of stakeholders in developing long-term strategies to enhance community awareness about CC; (2) creating mechanisms to strengthen support from physicians in the community; (3) meeting the needs of the underserved within a specialised health service for SA immigrant women. |
| McAvoy and Raza[48] (1991) | | | | |
| Leicester, UK | 737 randomly selected Asian women; 18–52 years who were not recorded on the central cytology computer as ever having had a cervical smear. n=578 (declined n=159). | Prospective cohort RCT study (blinded trial) (1) visited, shown a video, n=263; (2) visited, shown a leaflet and fact sheet, n=219; (3) posted a leaflet and fact sheet, n=131; (4) not contacted at all, n=124. | Knowledge of early intervention. | Only 6 (5%) of those not contacted and 14 (11%) of those sent leaflets had a smear test during the study. Health education interventions increased the uptake of cervical cytology among women in Leicester who had never been tested. Visits and videos were most effective. |
| Ornelas et al[74] (2017) | | | | |

**Table 4** Continued

| Region | Sample size and sampling frame | Intervention | Focus | Findings |
|---|---|---|---|---|
| Greater Seattle, Washington, USA | 40 SA women, 20 Karen–Burmese and 20 Nepali-Bhutanese; 21–58 years (mean age 35 years); living in the USA for 5 years on average. Most did not speak English well or at all (75%); 8 years average of education; 65% married. 73% had Pap test since arriving to the USA, 70% in last 3 years. | Presurvey and postsurvey. The two health educators recruited participants through personal contacts they had in their community, as well as referrals from community advisors and participants with whom they had completed data collection. A pilot study to evaluate the acceptability and efficacy of the 17 min videos provided in their native language. | Behavioural Model Changes in CC awareness, intention to be screened for CC, CC-related knowledge. | Nepali-Bhutanese were significantly more likely to have been screened than Karen–Burmese (90% vs 55%). Women showed significant increases in knowledge for all the individual items, as well as the mean composite knowledge scores (5.6 to 9.3, p<0.001) after viewing the video. There were also increase in knowledge for individual items across ethnic groups; however, not all were significant. Mean changes in the knowledge score were significant for women in each ethnic group (5.4 to 9.2, p<0.001 for Karen–Burmese and 5.8 to 9.5, p<0.001 for Nepali-Bhutanese). Women indicated high satisfaction with the video length and very few women reported about anything they did not like. |
| **Breast and cervical cancer** | | | | |
| Kernohan[72] (1996) | | | | |
| Bradford, UK | October 1991 to March 1993, a stratified sample of 1000 women (670 SA, 163 African-Caribbean, 96 Eastern European and 71 other). | Community development approach—preintervention–postintervention. Two health promotion facilitators undertook community development work in both formal and informal settings. Women were interviewed at the beginning of the project and 6 months after the health promotion intervention. | Knowledge about CC and BC. | SA women had the lowest levels of knowledge and also showed the most significant improvements. Significant increases in attendance for cervical smear and BC screening were self-reported. |
| Lofters et al[77] (2017) | | | | |
| Ontario, Canada | 624 phone calls made, of which 257 were to SA women. 129 (50%) of SA women spoken to directly by SA HAs. | Three quality improvement initiatives for four physicians using a snowballing technique: (1) educational vidoes shown in the waiting room and/or 1–1 education with patients by SA HAs; (2) 1–1 education with patients identified by SA HAs or physician; (3) phone calls to patients by SA HAs. | Transtheoretical model. Barriers and facilitators to cervical and mammography screening. | Most SA women spoken to by a SA HA indicated a willingness to get screened for BC or CC and some went on to action their screening intention. Making phone calls to patients to invite them for screening had the most reach and most appeal. The initiatives were reported to be resource intensive for physicians even with voluntary SA HAs involved. However, using SA HAs showed promise to increase awareness and willingness to be screened for cancer. |

BC, breast cancer; CC, cervical cancer; FGD, focus group discussions; HA, health ambassador; HBM, health belief model; HCP, healthcare provider or professional; IDI, in-depth Interviews; Pap, Papanicolaou; RCT, randomised controlled trial; SA, South Asian; SRH, sexual and reproductive health; CBE, clinical breast examination.

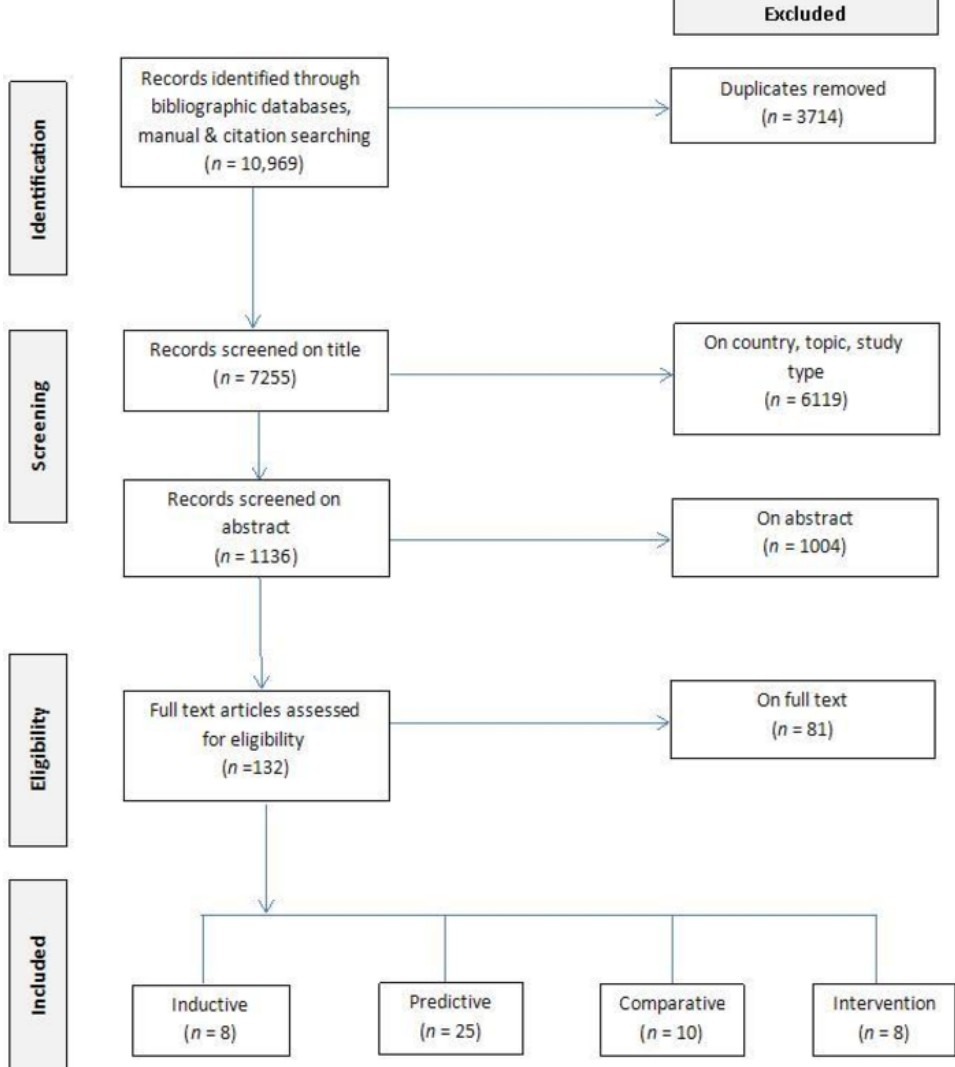

**Figure 1** PRISMA (Preferred Reporting Items for Systematic Reviews and Meta-Analyses) flow chart of the study selection process. Adapted from: Moher *et al*[38], *PLoS Medicine* (open access).

who were more acculturated to Western host countries, operationalised as time spent in those countries, were more likely to attend screening.

### Inductive studies

The eight inductive studies (table 1) were conducted in Canada and the USA, among Pakistani, Indian, Sri-Lankan Tamil and Bhutanese populations. Sample sizes ranged from 20 to 68, with a median of 43. Women had varying lengths of residency and were mostly born outside the host country. Six studies[41–46] used in-depth interviews and/or focus group discussions. One study[47] employed concept mapping using participatory research methods. Pons-Vigues *et al* acknowledged difficulties in interviewing women whose cultural backgrounds differed most from their own and championed the need for cultural intermediaries. However, intermediaries were not used in the other studies. Studies focused on the experiences of women themselves and did not interview family members or healthcare providers.

Data synthesis generated three overarching themes: '*Knowledge, attitudes, understanding of cancer and cancer prevention*'; '*Culture*'; and '*The process of Cultural adaptation*' to the host country.

### Knowledge, attitudes, understanding of cancer and cancer prevention

Neither cancer nor intimate body parts are commonly discussed in some South Asian cultures.[42 46] All studies showed that women lacked basic understandings of cancer, cancer prevention or early detection. Breast cancer was viewed by some women as a '*white woman's disease*'[41] that did not occur in their community.[45] Others considered cancer to be incurable and early detection and intervention futile.[45] Cervical cancer was often not known or understood. For example, some Bhutanese refugees in the USA had not heard of cervical cancer.[45] Indian Sikhs in Canada, living in a culture where sexual and reproductive health is rarely discussed, referred to the cervix as an 'unknown' and unspoken part of their

body.[46] Those aware of cervical cancer perceived the principal risk factors to be inseparable from those for general health, rarely mentioning the discrete risk factors of having multiple sexual partners, not using barrier contraception or screening.[45]

Studies attributed a lack of understanding of cancer to two main factors. First, religious fatalism meant that cancer was seen as predestined, as divine retribution for sins or as a dearth of moral character.[42] Second, all studies pointed to the curative focus of healthcare in countries of origin as a reason for some women's failure to understand the concept of prevention[42 48] and consequent belief that healthcare seeking is unnecessary in the absence of symptoms.[41 44 46 49]

## Culture

Family responsibilities were salient to women. This had three implications, one positive and two negative. First, women felt strong responsibilities to remain in good health and to protect family members from cancer.[40] In some cases, this facilitated screening attendance, however, some women found no time to attend screening due to family responsibilities.[44 49] This facilitated screening attendance. Second, notions of stigma precluded screening. Themes of 'shyness', 'modesty' and 'embarrassment' about revealing intimate body parts were important.[41] For example, Bhutanese refugees worried that attending a Pap test would damage reputations for chastity.[45] A Canadian study showed that women may be more likely to attend a cervical smear if the family doctor was female.[49] Both breast and cervical cancer were seen as stigmatising[46] and to some women this extended to screening.[42 45] Indeed, some Sri Lankan Tamils worried that attending a mammogram would lead people to think they already had breast cancer.[42] Third, women's behaviour was often subject to influence from male members of the family. Women frequently followed family advice for healthcare provided by males and elders, generally against screening, and felt the need to avoid conflict within the family associated with assertions of independence.[41 46] Another study showed that women felt family members to be supportive.[49]

## The process of cultural adaptation

South Asian nations have largely curative health systems, in which health costs are required to be paid by patients and there is no free access to healthcare. This contrasts with preventive healthcare models in host countries, and as with other health issues, South Asian women showed little understanding or orientation toward cancer prevention,[46] although this evolved with time as awareness of the culture of the host country increased.[43 47] Women appreciated healthcare professionals who understood and respected values of personal modesty/shyness.[45] South Asian women in Canada emphasised the value of being chaperoned to screening appointments that may have been located away from their local community, for assistance with language barriers, to alleviate feelings of personal vulnerability and to avoid being alone with doctors.[41]

## Deductive studies
### Study quality

Predictive and comparative studies contained similar limitations to quality. The first limitation was the poverty of theory. With the exception of Pons-Vigues and colleagues, whose deductive study[50] was informed by their earlier inductive work,[43] we noted little correspondence between inductive themes and hypotheses tested in the deductive research. Studies focused on knowledge of cancer and screening, but were not informed by themes of fatalism, non-understanding of preventive healthcare or cultural and family systems found in the qualitative research. Instead, studies were theoretically based on Western health behavioural theories, such as the HBM,[51] with limited applicability to South Asian populations. Similarly, the concept of acculturation was invoked in predictive studies, but was operationalised in a limited way, focusing on time spent in the emigrant country and language preferences. Other deductive work was not theoretically based.

Studies were also affected by methodological limitations. Three database linkage studies (from the same research team)[52–54] and two cluster sampling studies[55 56] provided samples with a potentially high degree of population representativeness, with random digit dialling techniques providing some confidence that samples may be representative.[57 58] Other studies used poor sampling techniques, including selection of South Asian names from phone directories or sampling at cultural events or other locations with high proportions of South Asian women, providing less confidence. One comparative study recruited a local population through random-digit dialling, but gathered a convenience sample of South Asian women through community centres and associations. This difference in sampling means reducing the value of the comparison between samples.[59] Definition of a South Asian population differed between studies, some examined women born in South Asia, others second-generation immigrants and some examined self-identified ethnicity.

It is important that attendance is recorded objectively.[60] All studies but the three linkage studies[52–54] used non-verified self-reported attendance and one used a hypothetical scenario of an offer to attend screening.[57] These outcomes included timely screening attendance (eg, previous screening was within a specified time period or reported regular timely testing) or whether women had ever been screened in the past.

### Predictive studies

It is strongly recommended that predictive studies be conducted prospectively to eliminate the problem of reverse causality.[60] All of the 23 predictive studies were cross-sectional and causal interpretation is difficult.

Lower screening rates were noted among women with no health insurance, younger women and women with lower levels of education. Studies did not provide consistent evidence that low knowledge predicted reduced likelihood of attendance. Lower knowledge was associated with a reduced likelihood of mammography screening

in two studies,[61 62] but did not predict the likelihood of hypothetical acceptance of a cervical screen.[57] Lower attendance was associated with a greater number of self-reported barriers to screening[61–65] although one study found the opposite.[66] However, the instruments used to assess barriers were largely based on existing instruments developed among Western samples that do not reflect South Asian concerns such as adapting to a new culture, language or health system.

Where acculturation was examined, less time spent in the host country was the strongest predictor of non-attendance, although one study cited lower preference for the host language (usually English) compared with women's native language[65] and another self-perceived poorer command of the host language.[66] Vahabi et al[52] found that South Asian women were less likely to attend mammography screening if their general practitioner (GP) had qualified outside the host country. Lofters et al[77] also found that South Asian women were less likely to attend mammography screening if their GP had qualified outside the host country. Vahabi and Lofters[52] showed benefits in mammography and cervical screening, respectively, for female family doctors.

## Comparative studies

Nine of the 10 comparative studies compared South Asian women with host populations, and eight compared South Asian women to other minority groups. South Asian samples often differed from comparison samples on demographic variables such as socioeconomic status (from lower socioeconomic backgrounds) and relationship status (mostly married), which limits trust that can be placed on comparisons if these factors are not statistically adjusted for.

Four comparisons with host populations showed South Asian women to have lower screening rates,[55 56 59 67] but two did not.[68 69] Of these studies, Dunn and Tan and Marlow et al used sampling techniques more likely to derive representative samples. Lower screening rates may be attributable to the knowledge deficits and greater perceived barriers observed in some studies.[50 56 70]

Two methodologically rigorous comparisons between South Asian and other minority groups[52 55] used population sampling and statistically adjusted for demographic differences between samples. Vahabi et al[52] also used an objectively verified indicator of mammography attendance. Both showed South Asian women to have lower attendance rates than other immigrant women. In two studies, Indian women had lower knowledge of cancer and screening than Chinese or Malays.[55 70] Pons-Vigues et al[50] and Teo et al[67] showed Indian and Pakistani women perceived fewer barriers arising from lack of knowledge about preventative screening than other immigrant groups and highlighted that many of the women thought that routine blood tests and urine tests would detect broader health issues such as cancer.[50] In another study,[68] Indian women perceived themselves to be less vulnerable to getting breast cancer, did not view breast cancer as a

serious illness and were more likely to claim that they did not know 'where to find a mammogram'.

## Intervention studies

Community educational programmes promoted breast and cervical cancer screening across the eight intervention studies. Four of the studies were precommunity and post-community based interventions,[71–74] two were randomised controlled trials,[48 75] one a time series study[76] and one a snowballing technique used as part of quality improvement initiatives for physicians.[77] Sampling was predominantly among South Asian women as a group, which eliminates comparisons between the different South Asian populations. Studies employed various methods of socioculturally tailored, language-specific health education materials and participants were recruited from primary care or South Asian community venues and residences. Recruitment was opportunistic via local newspapers, surveys conducted in community settings, South Asian nurses and link health workers. No study examined age trends,[73] and participants had met the researchers before which may constitute a bias.[48] Controlled studies were conducted in close-knit communities which may have led to intervention contamination into the control groups. Increased screening rates were reported for four studies but many were self-reported[71 72 77] or were indicated to improve,[74 77] rather than from objective indicators.[48] No long-term change in screening uptake was reported for five studies,[73–77] but they showed an increase in knowledge of breast cancer among South Asian immigrant women and reduced the misperception of short survival after diagnosis.

## DISCUSSION

Prominent across study types were the findings that South Asian women had poorer understandings of cancer and cancer prevention and that they perceived greater cultural and structural barriers to screening than host country women.

Lack of understanding by South Asian women about the need for asymptomatic screening has important ramifications. Predictive studies showed greater knowledge to be associated with screening attendance. The inductive research yielded some plausible reasons for this. Many women held fatalistic views or beliefs that cancer is incurable, while others believed that cancers could be identified in routine health testing. Others were unaware of the existence of cervical cancer in particular and did not perceive threat to themselves or their communities. The role of males was also important, with male family members sometimes negative about screening and women unwilling to provoke conflict within the family by attending. While there is a clear need to change such beliefs, the inductive studies showed this to be a challenging task for two reasons. First, understandings were embedded within religious and cultural traditions and cannot be addressed in isolation to those traditions. Thus, a simple educational intervention is likely to have limited effect. Accommodations will

need to be reached with communities that allow a creative integration of cancer awareness within existing belief structures. Second, some women were largely unaware of the concept of disease prevention. Thus, the promotion of specific cancer awareness and understandings are unlikely to be helpful until a wider understanding of prevention is reached.

Predictive studies showed the importance of perceived barriers (eg, lack of education, no health insurance, no family history, lower mammogram importance, less years living in host country, unmarried, language barriers, low self and outcome efficacy for screening), but these barriers pertained only to generic barriers faced by either all women or all immigrant women, irrespective of culture. Acculturation, in terms of time spent in the host country and mastery of the language was associated with increased screening likelihood, but these issues are likely to exist for all immigrant women and fail to reveal specifically South Asian issues. Inductive studies provided more subtle and culture-specific indications of the barriers perceived by women. Many were cultural. In particular, women spoke of the importance of female modesty and stigma associated with cancer that also affected willingness to be screened. While the importance of female testing staff from South Asian backgrounds and use of South Asian chaperones is emphasised, this cannot address the wider cultural issues of modesty and stigma. One finding that offers encouragement is that personal health is important to South Asian women because it helps them to care for their families.

Interventions will need to be conducted more widely than merely targeting women and their beliefs. Males occupy decision-making roles in some South Asian families and women may not wish to challenge this (see also Kinnaird et al[78 79] and Senarath and Gunawardena[78 79]). Thus, addressing the views of male family members and other community opinion leaders is also important.

## Limitations

The following limitations were identified within the review. First, many of the included studies were conducted in the USA, where screening services can require payment, which may not be comparable to other health services. Second, due to the small number of published studies, it is difficult to identify factors unique to groups of South Asian women based on nationality, geographical region or religion. By necessity, we discuss findings in terms of a generic 'South Asian' population, but are aware of variance between South Asian populations according to nationality, region, culture and religion. Finally, few studies used sampling techniques that are population representative, employing samples based around community activities. This may introduce unknown biases in findings associated with non-sampling of women who are less likely to attend such activities.

## Future research

Stratifying the analysis by study methodology brings two benefits: greater confidence can be placed on findings that transcend methodologies than those that are contained within one method, and studies with similar methodologies can be critiqued in ways appropriate to those methodologies. This review emphasises the generally poor quality of the deductive literature, which is problematic for developing epidemiological estimates of the prevalence of factors that inhibit or facilitate screening and the extent to which they do so. Such estimates would provide information pertaining to the relative importance of facilitators and inhibitors, and how changing them may influence screening attendance.[30] Failure to incorporate inductive findings into the design of deductive studies means that many inductive findings are untested in a population context. Further, deductive studies themselves used flawed designs as they were generally atheoretical or based on health behavioural models developed in Western populations and thus potentially lacking insight into South Asian issues. Translation of inductive findings to a deductive context will require the development of valid and reliable instruments to assess cultural understandings, beliefs, norms and practices.

There is room for well-designed operations research for interventions that target South Asian women who underuse and who have never been screened. These studies will also need to use better empirical methods. Few studies used sampling techniques that can be confidently claimed to be population-representative. Thus, there is a risk that South Asian people who attend community events, which was a common sampling strategy, are not representative of those who do not. It is important to employ best practice in study design for screening attendance research; the use of prospective predictive studies and objectively verified reporting of attendance from clinical records.[60] Adequate sampling frames need to be established. First, this involves a distinction between South Asian women as a minority group or as an immigrant group. The former can comprise women with high degrees of familiarity with the host country, but who nonetheless may be faced with cultural barriers deriving from their countries of origin. The latter group will reflect the problems of adjustment faced by recent immigrants. Studies will also need to use population-representative sampling techniques.

## Recommendations for practice

Findings from all study types demonstrate that interventions should be sensitive to cultural norms. In particular, studies emphasised the importance of language, female practitioners and the importance of community approval and involvement. Interventions at the community level will be necessary to surmount the cultural barriers identified in the inductive studies.

It is worrying that the findings indicated that younger women and women with lower levels of education were less likely to attend for screening. There is some evidence that South Asian women might experience breast cancer at an earlier age,[80] thus interventions may need to be targeted at educating South Asian women who are younger. Encouraging female family members to become more involved as chaperones and translators could also be helpful and may form a mechanism for educating young women simultaneously.

Information aimed at South Asian women who are invited for breast and cervical screening should highlight the presence of female practitioners and exclusively female environments at breast and cervical screening sites in the UK.[81] There is limited use of written communication in South Asian languages, although 70% of screening units across the UK want to provide information in patient's language.[82] This may help improve South Asian women's knowledge, make informed choice/consent, have better patient experience and eventually help in improving their screening uptake rates.

Interventions to increase uptake rates need to be long term, multifaceted and tailored to the specific needs of the local community by, for example, developing close links with the community through Health Education workers. South Asian community members, including males and opinion leaders, should be encouraged to be involved and coproduce engagement strategies within community settings. Reducing ethnic inequalities in uptake rates of breast cancer screening needs to remain a policy priority of breast screening programmes.

**Author affiliations**
[1]University of Liverpool, Liverpool, UK
[2]NIHR Collaboration for Leadership in Applied Health Research and Care, University of Liverpool, Liverpool, UK
[3]School of Natural Sciences and Psychology, Liverpool John Moores University, Liverpool, UK
[4]Royal Liverpool and Broadgreen Hospital NHS Trust, Liverpool, UK
[5]School of Health Sciences, University of Central Lancashire, Preston, UK
[6]The Nightingale Centre and Genesis Prevention Centre, University Hospital of South Manchester NHS Foundation Trust, Manchester, UK

**Acknowledgements**  The authors thank Professor Andy Clegg, the Royal Liverpool and Broadgreen Hospital Trust, the PREVENT Breast Cancer Charity in Manchester and the South Asian Women who contributed to the study at public engagement events. The authors also thank the public advisers and wish luck to them in any future research.

**Contributors**  PS and SLB designed this study; PS, SLB, RMAdC, DR, NT, SA, KB, AJ, MC and EK contributed in finalising search terms; RMAdC, EK and DR searched databases and RMAdC, DR and PS collected full-text papers; PS, SLB, RMAdC and DR extracted and analysed data; NT and SA contributed with reviewing abstracts, reading full text papers and extracting data. PS, RMAdC and SLB wrote the manuscript and KB reviewed the manuscript. Public advisers who contributed throughout this systematic review: NT and SA.

**Funding**  PS and SLB are part-funded by the National Institute for Health Research Collaboration for Leadership in Applied Health Research and Care North West Coast (NIHR CLAHRC NWC) (grant/award no: CLAHRC-NWC-015).

**Disclaimer**  The views expressed are those of the author(s) and not necessarily those of the NHS, the NIHR or the Department of Health and Social Care.

**Competing interests**  None declared.

**Patient consent**  Not required.

**Provenance and peer review**  Not commissioned; externally peer reviewed.

**Data sharing statement**  There are no unpublished data for this review.

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
