## [Reviewer comments · BMJ Open]

ARTICLE DETAILS

TITLE (PROVISIONAL)	A systematic review of barriers and enablers to South Asian women's attendance for asymptomatic screening of breast and cervical cancers in emigrant countries
AUTHORS	Anderson de Cuevas, Saini, Pooja; Rachel; Roberts, Deborah; Beaver, Kinta; Chandrashekar, Mysore; Jain, Anil; Kotas, Eleanor; Tahir, Naheed; Ahmed, Saiqa; Brown, Stephen

VERSION 1 – REVIEW

REVIEWER	Jennifer Hulme University of Toronto, Toronto, Canada
REVIEW RETURNED	28-Dec-2017

GENERAL COMMENTS	Overall a very interesting and timely paper. My main concerns are around the methods. A number of relevant studies were not captured in the review and it's unclear at which step they were eliminated (i.e. studies with poor methods were not eliminated, as per the methods). There is a lack of clarity on the added value of using this literature review methodology, although the explanation is almost there. The discussion fails to illustrate how their findings differ or are similar to other reviews. ABSTRACT: Page 3 Line 31: Canada doesn't have a payment oriented system. All screening is free of charge. And yet the same findings apply to South Asian populations. Page 3 Line 41: Why will high quality research using prospective cohorts, say, change the investments or outcomes of interest? I would challenge the notion that we don't understand the dynamics behind low screening rates. Introduction: Overall strong until the rationale starting on page 5 line 16. Would benefit from a balanced view. For example, breast cancer screening likely has little impact on mortality. Other than lower screening rates, Are there other reasons why South Asian women have worse outcomes? Higher rates of HPV, more invasive subtypes, ETOH, obesity - re breast cancer - or higher rates of BRCA1-2? Line 45: Is it really unclear why screening rates are lower? We have a fairly good idea of why. Examples from just one author: https://scholar.google.ca/citationsuser=djlzuBQAAAAJ https://www.ncbi.nlm.nih.gov/pubmed/27733161
--

Page 5 Line 33 Clarify why you are conducting an integrative review - i.e. to include experimental and non experimental research, and data from theoretical and empirical literature. How do you expect this to reduce bias, or change the recommendations or conclusion compared to Crawford et al and Sokal's reviews. i.e. how did they handle the 4 types of research used in migrant health? And why does it matter? I know you address this on p6 under aims of the review, but clarify this so the paragraphs are not 'floating' out of context. Use this space to clarify the objectives of the review.

Methods:

Selection criteria: did you study "attendance at screening", or also look at knowledge and attitudes and health beliefs around cervical cancer and breast cancer screening that did not actually 'study attendance'? This is important, as the review doesn't capture several interesting, relevant inductive studies that I have read, and I wonder if this is because of the inclusion criteria limiting inclusion, or were they excluded on the basis of poor quality. These studies go further than the simplistic predictive studies described in tis review to look at acculturation, language, and health system factors.

Results:

Page 14 line 36: same question. Is your research question about studies which measures screening attendance, or studies that aimed to understand the reasons why screening rates are low.

Discussion:

The Role of males occupying decisions- I'm not sure that the review reflected a trend that men are often a barrier in obtaining screening, although mobility outside the house is often limited.

The review failed to reveal some interesting trends in the literature. Canadian studies have shown that south asian women are much less likely to be screened if their family physician/ GP is also South Asian. Modesty and stigma might be addressed through a health system lens, where there are multiple ways by which all women, especially South Asian women, are brought into screening, and not just through the relationship with their GP.

Limitations: Again, there is no payment for health care in Canada. All of the Canadian studies found no barriers for women in terms of financial barriers, except for transport to get to the clinic.

Future Research: I agree that stratifying the studies by methodology brings a helpful organisation to the review.

Few studies used sampling techniques that are population representative because "population representation" may not be that important to informing culturally relevant and helpful interventions to address the screening gap.

Recommendations for practice: Here I really agree with the recommendations which I feel are really well supported in the literature at multiple levels. How do you rectify the recommendations for further large scale population-based studies, and the confidence with which you are able to make these recommendations? Perhaps what you are trying to say is that there is room for well designed operations research for interventions that

	target under and never screened South Asian women?
REVIEWER	Syeda Zakia Hossain University of Sydney, 75 East Street, NSW 2141, Australia
REVIEW RETURNED	09-Jan-2018
GENERAL COMMENTS	Reviewer's report: Manuscript title: A systematic review of barriers and enablers to South Asian women's attendance for asymptomatic screening of breast and cervical cancers in emigrant countries The manuscript is a systematic review of the literature on barriers and enablers to South Asian women's attendance in breast and cervical cancer screening living in emigrant countries. The aim of the systematic review was to identify the cultural, social, structural and behavioural factors that influence South women's attendance in asymptomatic breast and cervical cancer screening rates in order to improve screening rates and propose priorities for further research. The authors clearly stated the aims of the systematic review, discussed the process of selecting literature that falls under the broad topic under investigations. The authors' used search engine that is commonly used for identifying literature and used the existing databases appropriately. They also utilised integrative reviews process to compare findings of inductive, predictive, comparative and intervention studies, and also assessed consistencies between methods used in these studies efficiently. Methods of the systematic review clearly presented the steps use in selecting the literature including inclusion criteria. Also, the screening process of the literature to determine eligibility was clearly explained. Exclusion criteria were also stated but it is limited as it fails to provide justifications. Results: Results are presented under two broad methodological approaches, inductive (qualitative) and deductive (quantitative) research. The deductive research was based on three key approaches, such as predictive, comparative and intervention. The results from the inductive studies were presented under three broad themes. The findings showed that the South Asian women with more complete understandings of cancer and screening were more likely to attend the screening. While inductive study findings suggested a number of key factors that inhibit or enhance participation in the screening. However, factors such as family responsibilities were seen as an enabler to attend the screening, which is also shown an inhibitor in the participation in the screening in other studies. Women found no time to spend to attend the screening due to family responsibilities (Robinson, et al, 2016). The authors discussed scientifically some of the methodological limitations in the deductive studies including population representativeness of the sample, sample selection, confidence issue with samples, the definition of South Asian women and identifying South Asian women. It is stated that 'the definition of a South Asian population differed between studies, some examined

	women born in South Asia, others second-generation immigrants, and some examined self-identified ethnicity'. These inconsistencies could influence the findings of the study. Further, the authors noted that little correspondence between inductive themes and hypotheses tested in the deductive research exist. This is one of the key findings of the systematic review which will assist in the future research in this area. The researchers presented the deductive study results while discussing the quality of the study, and commented on the limitations of the study first before discussing the findings of these studies. It is important to discuss the contribution of these studies first, how well the methods were executed and results were presented followed by the limitations of the study. Below some of the minor errors need rectification:  1. Page 11, paragraph 1: Predictive studies showed that the number of perceived barriers inhibited screening, but did not provide more details on the barriers. It is important to add the barriers to screening attendance. 2. The results presented lacking some clarification, such as page 13, line 43: 'With the exception of Pons-Vigues and colleagues, whose deductive study 45 was informed by their earlier inductive work 39' it is unclear whether the authors referring to the reference list no 45 or list 39 or these are the study sample size. It needs clarification. 3. The authors proposed a number of interventions and suggested future work. They clearly stated that 'It is worrying that the findings indicated that younger women and women with lower levels of education were less likely to attend for screening. Interventions need to be targeted at educating South Asian women who are younger, not married and less educated'. However, it is not clear why young South Asian women should be targeted. It is clear from the existing literature that breast cancer happened to women above age 49 and therefore breast cancer screening is free for women 50-70 years in many developed countries including Australia. However, breast cancer happens to women in developing countries, in particular, those from South Asia at an early age such as age 35 and above. Therefore it is important for these women to be well educated on the importance of screening at an early age. These should have been explained when recommending why young South Asian women should be targeted. After rectifying the above concerns, the manuscript should be considered for publication.
--	--

VERSION 1 – AUTHOR RESPONSE

Editor comments:

Comment 1: Please ensure that you make the research question clearer particularly whether this is just about “knowledge and attitudes and health beliefs” about screening or whether it is also about attendance at screening, or both?

We have clarified in the abstract, aim (page 2) and selection criteria within the method (page 8) that the review is specifically about cultural, social, structural and behavioural factors that influence asymptomatic breast and cervical cancer screening attendance.

Comment 2: Please rephrase, in the introduction, “screening probably reduces mortality and morbidity through early detection and treatment”, as this is still controversial, with studies that find mortality reduction while others don’t.

We agree that the literature in this area is nuanced, and have changed the text to better represent the current state of the literature. In particular, we make a distinction between intention-to-treat research that examines population mortality, and the less commonly-used analysis of the effects of screening on individual mortality that excludes women who do not accept screening invitations. We have also addressed the important point made by Reviewer 1 that screening attendance is not the sole putative reason for higher mortality rates in South Asian women.

‘Whilst South Asian and host populations may differ over a range of factors that influence mortality, such as tumour sub-type and HPV status (Gomez 2010), one potential cause of greater mortality is that South Asian women show a lower likelihood of attending routine mammographic and Papanicolaou (Pap) screening. Screening is widely available in most high income countries 6-8. Some research shows population mortality benefits of screening programmes 10 11 , although other studies find no effect 9. Importantly, greater mortality benefits are found at the individual level, where studies confine analyses to women who accept screening rather than those who are merely invited (because some women decline screening) 12. Compared with the host population, South Asian women in England show lower uptake of breast screening services 13-16, particularly those from lower socioeconomic groups 13 17 18 and a higher proportion have never received cervical screening 19. This is also the case in the USA 20.’ (page 4)

Comment 3: Perhaps authors could discuss the evidence base better and convey a more “balanced view”, as reviewer Hulme says.

Reviewer 1’s comment refers to the evidence pertaining to screening effects on mortality. Please see our response to Comment 2.

Comment 4: Please discuss how does this review compare with others.

Previous reviews have combined studies that use different methodologies. We used an integrative approach that stratified our analysis by study method, and then integrated findings for different methods. We have updated our explanation of the benefits of this approach on page 5.

‘Crawford et al. and Sokal’s reviews have limitations. Both, combined studies with differing methodological approaches to achieve integrated descriptions of findings. This approach provides a comprehensive overview, but may lead to interpretation bias because it does not separate content from method 26. This leads to two limitations. First, critical examination of study quality is more difficult when varying methodologies are used. Thus, the value of findings cannot be easily moderated or weighted by quality appraisal of the reviewed studies. Second, it is important that findings are replicated across methods. For example, inductive research permits detailed phenomenological understandings of factors that facilitate or inhibit screening, but not epidemiological estimates of the prevalence of these factors or the magnitude of their influence on screening. This requires well-designed quantitative studies³⁰. Similarly, quantitative research alone is unlikely to be sensitive to local complexities unless complemented by inductive approaches. When these approaches are conflated, as with Crawford et al. and Sokal’s approach, the reader cannot determine if insights are or are not replicated across different approaches’ (page 5)

This point is later referenced in the discussion, where we voice concern about the problems of deductive approaches and suggest ways of better conducting this research:

'This review emphasises the generally poor quality of the deductive literature, which is problematic for developing epidemiological estimates of the prevalence of factors that inhibit or facilitate screening and the extent to which they do so. Such estimates would provide information pertaining to the relative importance of facilitators and inhibitors, and how changing them may influence screening attendance.' (page 22)

Comment 5: Please complete and include a PRISMA checklist, ensuring that all points are included and state the page numbers where each item can be found.

The PRISMA checklist has been completed and the page numbers added. This has been uploaded as a supplementary document

Comment 6: Please include the dates of the search in the abstract.

The dates of the search have been added to the abstract (page 2).

Comment 7: Your search only goes till July 2016 at the moment - please update this, if possible.

We have completed an updated search from 1st July 2016 to 23rd January 2018 and introduced seven new papers to the analysis. The manuscript has been changed to add insights from these papers. All four tables have been updated (pages 31-49). New references have been added (pages 26-30).

Reviewer 1 comments:

My main concerns are around the methods. A number of relevant studies were not captured in the review and it's unclear at which step they were eliminated (i.e studies with poor methods were not eliminated, as per the methods). There is a lack of clarity on the added value of using this literature review methodology, although the explanation is almost there. The discussion fails to illustrate how their findings differ or are similar to other reviews.

We have reconducted the literature search and randomly checked the results to detect whether we have systematically excluded studies during initial selection. We have also checked our review against those of Crawford et al and Sokol. We are not aware of omitted studies. Perhaps the studies to which the reviewer refers do not explicitly cover screening attendance or were published post July 2016. If so, we have included seven new studies in the review.

The reviewer makes the important point that our study should add value to previous work. As we detail in our answer to Comment 4 of the editor, we see cross-method concordance, or lack thereof, to be important. In particular, our study shows the weakness of the deductive research that we believe should follow-up the inductive findings. In the discussion, we explicate this by pointing out that, as Reviewer 1 suggests, we have information that allows us to conduct culturally sensitive programmes. However, due to the dearth of good deductive studies, we currently lack information on the prevalence and effect magnitude of factors that inhibit or facilitate attendance in South Asian communities.

Abstract:

Comment 1: Page 3 Line 31: Canada doesn't have a payment oriented system. All screening is free of charge. And yet the same findings apply to South Asian populations.

Following this comment we have updated the text within the paper and the points on page 3 have been updated:

'Many of the included studies were conducted in the USA, where screening services can require payment, which may not be comparable to other health services.' (page 21)

Comment 2: Page 3 Line 41: Why will high quality research using prospective cohorts, say, change the investments or outcomes of interest? I would challenge the notion that we don't understand the dynamics behind low screening rates.

As our responses to the editor's Comment 4 and the reviewer's first comment state, we see the establishment of a strong quantitative literature as important in helping us to understand the magnitude of the public health problem represented by cultural, social, structural and behavioural barriers to screening attendance in South Asian women. Whilst we fully agree that inductive approaches provide an understanding of the dynamics behind low screening rates, we see quantitative estimation of the prevalence and effects of these barriers as essential; not just in designing interventions, but in convincing policy-makers to invest in programmes. Certainly, good deductive research is important to attract funding to South Asian programmes in the UK's competitive funding environment.

Introduction:

Comment 3: Overall strong until the rational starting on page 5 line 16.

Would benefit from a balanced view. For example, breast cancer screening likely has little impact on mortality. Other than lower screening rates, are there other reasons why south Asian women have worse outcomes? Higher rates of HPV, more invasive subtypes, ETOH, obesity - re breast cancer - or higher rates of BRCA1-2?

Please see our response to the editor's Comment 2.

Comment 4: Line 45: Is it really unclear why screening rates are lower? We have a fairly good idea of why. Examples from just one author: <https://scholar.google.ca/citationsuser=djIzuBQAAAAJ>
<https://www.ncbi.nlm.nih.gov/pubmed/27733161>

We agree that the term 'unclear' implies that we have little understanding of the problem of lower screening attendance by South Asian women. This is not the case, and we have deleted the sentence. However, as we argue in response to the editor's Comment 4, high quality deductive evidence is important for policy and programme development.

Comment 5: Page 5 Line 33 Clarify why you are conducting an integrative review - i.e. to include experimental and non-experimental research, and data from theoretical and empirical literature. How do you expect this to reduce bias, or change the recommendations or conclusion compared to Crawford et al and Sokal's reviews. i.e. how did they handle the 4 types of research used in migrant health? And why does it matter? I know you address this on p6 under aims of the review, but clarify this so the paragraphs are not 'floating' out of context. Use this space to clarify the objectives of the review.

Thanks for pointing out the need to link these paragraphs to the previous content. We have done so in the text that we used to address the editor's Comment 4.

Methods:

Comment 6: Selection criteria: did you study "attendance at screening", or also look at knowledge and attitudes and health beliefs around cervical cancer and breast cancer screening that did not actually 'study attendance'? This is important, as the review doesn't capture several interesting, relevant inductive studies that I have read, and I wonder if this is because of the inclusion criteria limiting inclusion, or were they excluded on the basis of poor quality. These studies go further than the simplistic predictive studies described in this review to look at acculturation, language, and health system factors.

Where we examined predictive studies, we used attendance at screening as the key dependent variable. We eliminated a small number of studies because they did not measure this. With regard to the inductive studies, we selected studies that examined women's views about why they would or would not attend screening. Quality was generally not a basis for exclusion. Thus, we are concerned that the reviewer feels that there are relevant studies we did not capture. One possible reason is that we excluded studies where South Asian samples were conflated with other immigrant groups to the point where it would be unclear whether findings pertained to South Asian women. Another reason may be the cut-off date that excluded recent work. The search has now been updated.

Results:

Comment 7: Page 14 line 36: same question. Is your research question about studies which measures screening attendance, or studies that aimed to understand the reasons why screening rates are low.

Please see the previous response

Discussion:

Comment 8: The Role of males occupying decisions- I'm not sure that the review reflected a trend that men are often a barrier in obtaining screening, although mobility outside the house is often limited.

To emphasise this important point, we have added the following text to the discussion:

'The role of males was also important, with male family members sometimes negative about screening and women unwilling to provoke conflict within the family by attending.' (Page 16).

Comment 9: Limitations: Again, there is no payment for health care in Canada. All of the Canadian studies found no barriers for women in terms of financial barriers, except for transport to get to the clinic.

We have addressed the reviewer's comments and amended the text by pointing out that this limitation applies to US studies:

'The following limitations were identified within the review. First, most of the studies were primarily conducted in the USA. Findings from the US healthcare system, with payment-oriented healthcare, is not necessarily replicable in other health systems.' (page 22).

Comment 10: Few studies used sampling techniques that are population representative because "population representation" may not be that important to informing culturally relevant and helpful interventions to address the screening gap.

Following our argument about the importance of deductive research, it is of course important that deductive research is done well. Population representative sampling, or the closest that studies can realistically approximate to population representation, is important to successful deductive research.

Comment 12: Recommendations for practice: Here I really agree with the recommendations which I feel are really well supported in the literature at multiple levels. How do you rectify the recommendations for further large-scale population-based studies, and the confidence with which you are able to make these recommendations? Perhaps what you are trying to say is that there is room for well-designed operations research for interventions that target under and never screened South Asian women?

This relates to the previous points that we have made about quantification of factors that influence screening. In our view, this creates a need for the studies that we advocate.

Reviewer 2 comments:

Comment 1: Exclusion criteria were also stated but it is limited as it fails to provide justifications.

We have now included justification for our inclusion and exclusion criteria:

'The review included studies on the cultural, social, structural and behavioural factors that influence asymptomatic breast and cervical cancer screening attendance and cervical smear testing (Papanicolaou test) in South Asian populations. It was confined to host countries where mass screening programmes are available to the general public, including South Asian Women. The populations of interest were Bangladeshi, Indian, Pakistani, Sri Lankan, Bhutanese, Maldivian and Nepali populations (or ethnic subgroups thereof). To ensure that content was not confounded by inclusion of other groups, studies needed to report on samples or subsamples identifiable as wholly South Asian, meaning that we accepted papers that examined South Asian and other samples provided that authors explicitly specified where South Asian content differed from other samples (inductive studies) or where South Asian samples were analysed separately or were specifically identified in moderation analyses (predictive, comparative or intervention studies).

To ensure that the studies pertained to screening attendance, we excluded those that did not specifically refer to screening. Thus, studies solely covering general attitudes to breast or cervical cancer were excluded. The review did not include breast self-examination, diagnostic screening or visual or tactile examinations by healthcare professionals. We excluded studies of women in known high-risk groups who were engaged in monitoring programmes for genetic risk factors, hereditary breast and ovarian cancer syndrome, premenopausal or familial breast cancer. We excluded homogenous samples restricted to particular demographic groups because these are not population representative (e.g. a study of dental students).' (Pages 10-11).

Comment 2: Women found no time to spend to attend the screening due to family responsibilities (Robinson, et al, 2016).

We agree with the reviewers comments and have amended the text to include the Robinson reference:

'Family responsibilities were salient to women. This had three implications, one positive and two negative. First, women felt strong responsibilities to remain in good health and to protect family

members from cancer 40. In some cases this facilitated screening attendance, however, some women found no time to attend screening due to family responsibilities 44 49(page 14).'

Comment 3: Page 11, paragraph 1: Predictive studies showed that the number of perceived barriers inhibited screening, but did not provide more details on the barriers. It is important to add the barriers to screening attendance.

We agree with the reviewers comments and have amended the text:

'Predictive studies showed the importance of perceived barriers (e.g. lack of education, no health insurance, no family history, lower mammogram importance, less years living in host country, unmarried, language barriers, low self and outcome efficacy for screening), but these barriers pertained only to generic barriers faced by all women, irrespective of culture.' (pages 21-22)

Comment 5: The results presented lacking some clarification, such as page 13, line 43: 'With the exception of Pons-Vigues and colleagues, whose deductive study 45 was informed by their earlier inductive work 39' it is unclear whether the authors referring to the reference list no 45 or list 39 or these are the study sample size. It needs clarification.

We agree with the reviewers comments and have amended the text to make it clear that the numbers refer to the reference list:

'With the exception of Pons-Vigues and colleagues, whose deductive study⁵⁰ was informed by their earlier inductive work⁴³, we noted little correspondence between inductive themes and hypotheses tested in the deductive research. Studies focussed on knowledge of cancer and screening, but were not informed by themes of fatalism, non-understanding of preventive healthcare, or cultural and family systems found in the qualitative research.' (pages 15-16)

Comment 6: The authors proposed a number of interventions and suggested future work. They clearly stated that 'It is worrying that the findings indicated that younger women and women with lower levels of education were less likely to attend for screening. Interventions need to be targeted at educating South Asian women who are younger, not married and less educated'. However, it is not clear why young South Asian women should be targeted. It is clear from the existing literature that breast cancer happened to women above age 49 and therefore breast cancer screening is free for women 50-70 years in many developed countries including Australia. However, breast cancer happens to women in developing countries, in particular, those from South Asia at an early age such as age 35 and above. Therefore it is important for these women to be well educated on the importance of screening at an early age. These should have been explained when recommending why young South Asian women should be targeted.

We agree with the reviewers comments and have amended the text:

'It is worrying that the findings indicated that younger women and women with lower levels of education were less likely to attend for screening. There is some evidence that South Asian women might experience breast cancer at an earlier age⁸⁰, thus interventions may need to be targeted at educating South Asian women who are younger. Encouraging female family members to become more involved as chaperones and translators could also be helpful, and may form a mechanism for educating young women simultaneously.' (page 23)